# Real-Time Wildfire Detection Algorithm Based on VIIRS Fire Product and Himawari-8 Data

**Da Zhang** [1,2,3,4], **Chunlin Huang** [2,*], **Juan Gu** [5], **Jinliang Hou** [2], **Ying Zhang** [2], **Weixiao Han** [2], **Peng Dou** [2] **and Yaya Feng** [2]

1   Faculty of Geomatics, Lanzhou Jiaotong University, Lanzhou 730000, China; 11200879@stu.lzjtu.edu.cn
2   Key Laboratory of Remote Sensing of Gansu Province, Heihe Remote Sensing Experimental Research Station, Northwest Institute of Eco-Environment and Resources, Chinese Academy of Sciences, Lanzhou 730000, China; jlhours@lzb.ac.cn (J.H.); zhang_y@lzb.ac.cn (Y.Z.); weixiaohan@lzb.ac.cn (W.H.); doupeng@nieer.ac.cn (P.D.); fengyaya@lzb.ac.cn (Y.F.)
3   Gansu Provincial Engineering Laboratory for National Geographic State Monitoring, Lanzhou 730000, China
4   National-Local Joint Engineering Research Center of Technologies and Applications for National Geographic State Monitoring, Lanzhou 730000, China
5   Key Laboratory of Western China's Environmental Systems, Ministry of Education, Lanzhou University, Lanzhou 730000, China; gujuan@lzu.edu.cn
*   Correspondence: huangcl@lzb.ac.cn

**Abstract:** Wildfires have a significant impact on the atmosphere, terrestrial ecosystems, and society. Real-time monitoring of wildfire locations is crucial in fighting wildfires and reducing human casualties and property damage. Geostationary satellites offer the advantage of high temporal resolution and are gradually being used for real-time fire detection. In this study, we constructed a fire label dataset using the stable VNP14IMG fire product and used the random forest (RF) model for fire detection based on Himawari-8 multiband data. The band calculation features related brightness temperature, spatial features, and auxiliary data as input used in this framework for model training. We also used a recursive feature elimination method to evaluate the impact of these features on model accuracy and to exclude redundant features. The daytime and nighttime RF models (RF-D/RF-N) are separately constructed to analyze their applicability. Finally, we extensively evaluated the model performance by comparing them with the Japan Aerospace Exploration Agency (JAXA) wildfire product. The RF models exhibited higher accuracy, with recall and precision rates of 95.62% and 59%, respectively, and the recall rate for small fires was 19.44% higher than that of the JAXA wildfire product. Adding band calculation features and spatial features, as well as feature selection, effectively reduced the overfitting and improved the model's generalization ability. The RF-D model had higher fire detection accuracy than the RF-N model. Omission errors and commission errors were mainly concentrated in the adjacent pixels of the fire clusters. In conclusion, our VIIRS fire product and Himawari-8 data-based fire detection model can monitor the fire location in real time and has excellent detection capability for small fires, making it highly significant for fire detection.

**Keywords:** fire detection; VIIRS; Himawari-8; random forest

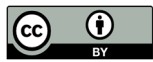

## 1. Introduction

Wildfires are major natural hazards that negatively affect human safety, natural ecosystems, and wildlife [1]. Although statistics show that the frequency and total burned area of global wildfires are decreasing year by year, some regions will experience larger and more intense wildfires [2,3]. Since September 2019, long-lasting, difficult-to-extinguish wildfires have erupted in many parts of Australia, covering a total area of more than 6 million hectares [4].

The timely and accurate monitoring of the location of wildfire occurrence plays an important role in wildfire suppression and reducing human casualties and property damage [5]. In recent years, remote-sensing satellite monitoring has gradually become an important tool for wildfire monitoring with its fast imaging cycle and wide coverage. This form of Earth observation is based on the detection of the characteristics of electromagnetic radiation (mainly infrared) that is emitted during the combustion of biomass. Currently, the absolute threshold method [6,7] and the contextual method [8] are widely used for satellite-based fire detection. The absolute threshold method was proposed by Flannigan et al. [6] in 1986. This method identifies fire pixels by comparing the brightness temperature of the pixel channel to be processed with a set threshold value. This method has good performance in wildfire monitoring in North American forest areas. The accuracy of the absolute threshold method, however, can be affected by environmental differences in different regions [7]. The contextual approach was proposed by Flasse et al. [8]. It compares the information of a potential fire pixel with that of its neighbors, and if their difference is greater than a set threshold, the pixel is identified as a fire pixel. The contextual method has wider applicability. Moderate Resolution Imaging Spectroradiometer (MODIS) thermal anomaly products based on the contextual approach have been evaluated in a large number of studies and have become standard for satellite-based fire detection [9,10]. The Visible-Infrared Imaging Radiometer Suite (VIIRS) active fire detection algorithm is an implementation of the MODIS thermal anomaly detection algorithm on the VIIRS sensor, which has a higher spatial resolution (375 m) [11]. A study by Oliva et al. [12] showed that VIIRS was able to detect 100% of wildfires larger than 100 ha in northern Australia. Although some fire-monitoring methods applied on polar-orbiting satellites have achieved satisfactory accuracy, the lack of temporal resolution of these satellites has made it difficult to achieve near real-time fire detection.

The Himawari-8 satellite is a new generation of synoptic meteorological satellite launched by Japan on 7 October 2014. This satellite has greatly improved temporal and spatial resolution compared with existing geosynchronous meteorological satellites, with temporal resolution up to 10 min and spatial resolution up to 0.5 km. In recent years, many scholars have used Himawari-8 to conduct research related to fire detection. Xu et al. [5] monitored the Esperance, Western Australia, wildfire in real time based on a spatially fixed threshold method. Experiments demonstrated that Himawari-8 had a strong immunity to smoke and thin clouds, was sensitive to small fires, and could provide valuable real-time fire information for wildfire management. Hally et al. [13,14] developed a daily temperature cycling model to detect anomalies in an image time series. This method, however, required a longer time series of cloud-free image elements as training data, which has limited its application. Wickramasinghe et al. [15] used middle-infrared (MIR) (2 km), near-infrared (NIR) (1 km), and RED (500 m) data from Himawari-8 to achieve continuous tracking of 500 m resolution fire lines. However, RED data are not available at night, resulting in the method being applicable only for daytime fire line monitoring. Although these methods are effective in fire detection, they do not take full advantage of the band and spatial information of the Himawari-8 data, thus also limiting fire detection.

In recent years, some scholars have used machine learning (ML) methods for fire detection. Chen et al. [16] combined long-time information with the gradient boost decision tree model to detect wildfires in the Yunnan region, and the accuracy of the algorithm was greatly improved. Jang et al. [17] used the random forest (RF) algorithm to detect Korean fires, and the algorithm had a great improvement in accuracy compared with other algorithms and had higher detection accuracy for small fires. Ding et al. [18] applied over 5000 images obtained from the geostationary Himawari-8 satellite of a severe Australian wildfire that occurred from November 2019 to February 2020 to train and test a fully connected convolutional neural network (CNN) for identifying the location and intensity of wildfires. The proposed CNN model obtains a detection accuracy greater than 80%. Kang et al. [19] used a CNN model to detect forest fires. The model accuracy was improved by adding spatial features and temporal features. Although these methods have achieved high

accuracy, the training dataset of these models is based on manual visual inspection or is obtained using actual surveys. The datasets constructed by these methods have limitations. Hassini et al. [20] discussed the case of simple visual inspection. This approach usually has low effectiveness because only fires with a large contrast between the pixels affected by the fires and the surrounding pixels can be visually identified on MIR images. It is not possible to observe small or starting wildfires, and only very large wildfires can be observed in plumes at the low spatial resolution of geostationary sensors. The use of recorded information enables a posteriori validation, in which the results depend strictly on the completeness and correctness of the catalogs, as well as on the minimum size of the recorded events that may vary in space and time according to the individual country's policy [21]. The lack of suitable datasets has limited the application of ML methods to active fire detection [22]. Therefore, it is crucial to create an automatically generated and representative training dataset for ML models.

Fine-resolution fire products are often used to verify the accuracy of coarse-resolution fire detection results and also can be used as a training dataset for fire prediction and detection ML models [23–25]. Zhang et al. [25] constructed a Himawari-8 fire label dataset based on high-confidence fire pixels from the VIIRS Fire product for training a daytime fire detection RF model. Although the model achieved good fire detection accuracy in Southwestern China, the representativeness of the fire label dataset constructed using only high-confidence fire pixels from the VIIRS Fire product needs further study. The response of different resolution sensors to the same fire may be different, which is often discussed when active fire detection results from coarser-resolution satellite data are evaluated using finer-resolution data. For example, the 30 m Advanced Spaceborne Thermal Emission and Reflection Radiometer (ASTER) onboard the National Aeronautics and Space Administration (NASA) Terra satellite was successfully used to detect active fires to validate fire detections from MODIS and the previous Geostationary Operational Environmental Satellite [26–30]. A logistic regression model can be used to estimate the probability of coarse-resolution fire detection in relation to the number of fine-resolution fire pixels contained within the coarse pixel. These studies all have shown a common pattern that the fire detection probability generally increases as the number of fine fire pixels increases within a coarse pixel. Therefore, in this study, we also considered this response characteristic when fine-resolution fire pixels are used to construct the coarse-resolution training dataset, with the aim of seeking suitable representative fire pixels.

Although machine-learning-based fire detection methods can reduce the generalization problems of simple threshold methods based on contextual statistical analysis, the fire label dataset used for machine learning (ML) model training is typically based on manual interpretation or field surveys, which are often labor-intensive and subject to human error. In this study, we used RF models for fire detection based on Himawari-8 data and constructed a fire label dataset using the stable VNP14IMG fire product. We addressed the challenge of constructing a coarse-resolution fire label dataset from fine-resolution fire products. We added band calculation features, spatial features, and auxiliary data for model training and evaluated the impact of these features on model accuracy. A feature selection process was used to exclude redundant features. We also constructed separate daytime and nighttime RF models to analyze their applicability. Finally, we extensively evaluated the model results by comparing them with the Japan Aerospace Exploration Agency (JAXA) wildfire product.

Section 2 provides an overview of the study area and the data used in this study, as well as the pre-processing of the data. Section 3 describes the detailed methodology of fire detection. Section 4 presents the fire detection results. Section 5 discusses the representative problems in constructing the training dataset, the advantages of the feature selection procedure in classification applications, and the reasons why the model produces omissions and commission detection. Section 6 concludes the study.

## 2. Study Area and Data

### 2.1. Study Area

Australia is one of the most wildfire-prone countries in the world [31], and southeastern Australia is considered a rare wildfire area, where fires occur mainly in the summer, as well as in the spring and autumn [32]. The southeastern part of Australia, which includes the states of New South Wales and Victoria and has about three-fifths of the total population of Australia, has an undulating terrain. In the west is a vast desert, and in the east is the Great Dividing Range, which runs from north to south through southeastern Australia. Forests along the Great Dividing Range are narrowly distributed, with approximately 28.4 million hectares of woodland, accounting for about 18% of the country's forest area, of which the dominant species is eucalyptus, which has a high oil content and is flammable [4]. Wildfires occur in the region throughout the year because of rainfall patterns and extreme weather. In the 2019–2020 wildfire season, bush and grassland fires in New South Wales reportedly burned more than 5.5 million ha, destroyed nearly 2500 houses, and killed 26 people. We selected southeastern Australia (shown in Figure 1) as the subject for this study.

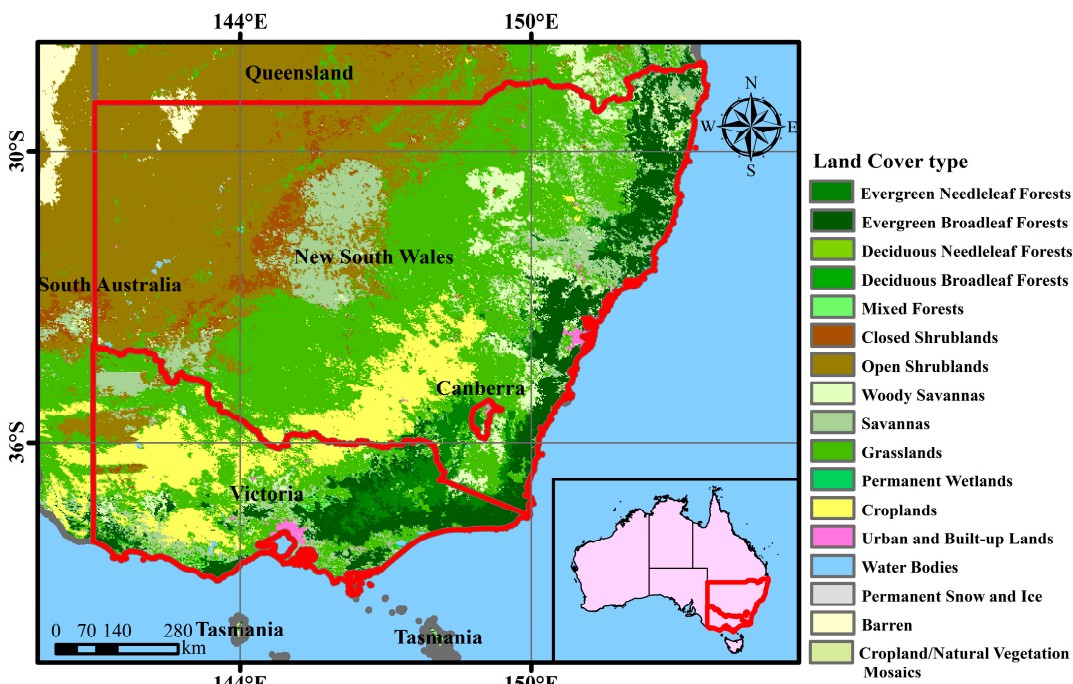

**Figure 1.** The spatial extent of the study area. The land cover is taken from MODIS 500 m Land Use/Land Cover (LULC) production for 2020.

### 2.2. VNP14IMG Fire Product

VNP14IMG is a fire product of the VIIRS sensor carried by the Suomi-National Polar-Orbiting Partnership (S-NPP) satellite. The product provides the latitude, longitude, time, confidence, and relevant band information of the fire pixel. This product was developed based on the MODIS thermal anomaly algorithm and can achieve a high spatial resolution of 375 m. Studies [11,33,34] have shown that the 375 m VIIRS fire product can easily detect small wildfires that are not detectable by MODIS. The product is available from the Fire Information for Resource Management System (FIRMS) (https://firms.modaps.eosdis.nasa.gov/download/)(accessed on 10 March 2023). FIRMS is part of NASA's Land, Atmosphere Near Real-Time Capability program for Earth Observation System and can provide search ranges in terms of the country and time historical VNP14IMG fire product. In this study, we selected the VNP14IMG fire product, which occurred in Australia in 2019 and 2020, and extracted fire pixels from the study area in

November and December 2019 and January 2020 according to time and latitude and longitude, Acquire_date, and Acquire_time information. Fire pixels with high and nominal confidence levels were used to construct the Himawari-8 label dataset for fire detection model construction and accuracy assessment.

### 2.3. Himawari-8 L1 Grid Data

Himawari-8 is the first of the third generation of geostationary weather satellites carrying the new Advanced Himawari Imager (AHI) instrument [35]. AHI can achieve data with a high spatial resolution ranging from 0.5 to 2 km and a high temporal resolution ranging from 2.5 to 10 min [36]. Himawari-8 L1 grid data are a standard product of AHI, with a spatial resolution of 0.02° covering the 60S–60N, 80E–160W region and allowing for full coverage of the study area every 10 min. Data can be obtained from JAXA's P-Tree system (ftp://ftp.ptree.jaxa.jp/jma/netcdf)(accessed on 10 March 2023). The product band information includes band 1–6 albedo data, band 7–16 bright temperature data (Tbb07–Tbb16), and attribute data, including satellite zenith angle, satellite azimuth angle, solar zenith angle, solar azimuth angle, and observation time. The band information of the product used in this study is shown in Table 1. We acquired Himawari-8 L1 grid data for November and December 2019 and January 2020 and cropped it to the study area.

**Table 1.** Himawari-8 AHI bands used in this study.

| Himawari-8 AHI Band | Bandwidth (μm) | Central Wavelength (μm) | Spatial Resolution (km) | Purpose |
|---|---|---|---|---|
| 3 | 0.03 | 0.64 | 2 | Cloud Mask |
| 4 | 0.02 | 0.86 | 2 | Cloud Mask |
| 6 | 0.02 | 2.26 | 2 | Water Mask |
| 7 | 0.22 | 3.85 | 2 | Fire detection |
| 8 | 0.37 | 6.25 | 2 | Fire detection |
| 9 | 0.12 | 6.95 | 2 | Fire detection |
| 10 | 0.17 | 7.35 | 2 | Fire detection |
| 11 | 0.32 | 8.60 | 2 | Fire detection |
| 12 | 0.18 | 9.63 | 2 | Fire detection |
| 13 | 0.30 | 10.45 | 2 | Fire detection |
| 14 | 0.20 | 11.20 | 2 | Fire detection |
| 15 | 0.30 | 12.35 | 2 | Fire detection |
| 16 | 0.20 | 13.30 | 2 | Fire detection |

### 2.4. Himawari-8 L2WLF Product

Himawari-8 L2WLF is a fire product released by the Japan Space Development Agency (ftp://ftp.ptree.jaxa.jp/pub/himawari/L2/WLF)(accessed on 10 March 2023), with a temporal resolution of 10 min and a spatial resolution of 0.02°. The product is based on the latitude and longitude information of the center of the fire pixel obtained from the MIR (3.9 μm) and thermal infrared (10.8 μm) band threshold tests of the Himawari-8 image. Data include the location of the fire pixels, the fire radiated power, and the band information of the fire pixels and can be used as the comparison data to verify the fire detection accuracy in this study.

### 2.5. MCD12Q1.006 Land Use Product

MCD12Q1.006 [37] is a MODIS land cover product used to determine the land cover type of the burn area, removing false alarms located in other land types. Figure 1 shows the results of the International Geosphere–Biosphere Programme scheme land classification for the MCD12Q1.006 data for the 2020 study area. The scheme contains 17 categories. In this study, we selected categories 1–10 and 14 as the underlying surface types where wildfires would be likely to occur. The data can be downloaded from GEE (dataset ID: ee.ImageCollection ("MODIS/006/MCD12Q1")). The original spatial resolution of the

MCD12Q1.006 data was 500 m, and it was resampled to 2 km using majority filtering to match the AHI spatial resolution.

## 3. Method

We split the goal of identifying fire pixels into two tasks. The first was non-fire pixels exclusion, which included cloud and water masks, and potential fire detection. The second task was to extract fire pixels from potential fire pixels based on the RF classification algorithm. The specific methodology for fire detection and the performance evaluation metrics of the classifier defined in this study are as follows.

### 3.1. Cloud and Water Masks

Cloud masks are a crucial part of any active fire detection [38]. Reflections from clouds in the MIR band can be mistakenly detected as fire pixels. Further, solar flares from water bodies can be misidentified as fire pixels [39]. Cloud masks and water masks can reduce the false fire detection caused by clouds and water. In this study, we referred to the method of Xu et al. [5] to detect clouds and water in the Himawari-8 images. A pixel satisfying the following equation is defined as a cloud pixel.

The daytime cloud pixel is shown in Equation (1):

$$(\rho_3 + \rho_4 < 0.9) \; and \; (T_{15} > 265) \; and \; ((\rho_3 + \rho_4 < 0.7) \; or \; T_{15} > 285) \,. \tag{1}$$

The nighttime cloud pixel is shown in Equation (2):

$$T_{15} > 265 \;, \tag{2}$$

where $\rho_3$ and $\rho_4$ are the reflectance of Himawari-8 band 3 and band 4, respectively, and $T_{15}$ is the brightness temperature of band 15. A pixel satisfying Equation (3) is defined as a water pixel:

$$A_6 < 0.05 \,, \tag{3}$$

where $A_6$ is the Himawari-8 Band6 albedo. Water masking at night was not required because of the low false positives produced by water bodies at night. We defined a nighttime pixel as a pixel with a solar zenith angle of more than 85°.

### 3.2. Potential Fire Detection

We used a preliminary classification to eliminate obvious non-fire pixels. Those pixels that remain, named potential fire pixels, are considered in the RF test to determine if they do in fact contain active fire. According to the method of Wooster et al. [39], the Himawari-8 pixel satisfying Equations (4) and (5) is defined as a potential fire pixel:

$$BT_{3.9} > C_{11}\theta_s + C_{12} \,, \tag{4}$$

$$BT_{3.9} - BT_{10.8} > C_{21}\theta_s + C_{22} \,, \tag{5}$$

where $BT_{3.9}$, $BT_{10.8}$ are the bright temperature values of band 7 and band 14 of the Himawari-8 pixel, respectively; $\theta_s$ is the solar zenith angle; and $C_{11}$ (−0.3 and 0.0), $C_{12}$ (310.5 and 280 K), $C_{21}$ (−0.0049 and 0.0), and $C_{22}$ (1.75 and 0.0 K) are the constants applied when $\theta_s > 85°$ and $\theta_s < 85°$. Because a key requirement for an enhanced fire detection algorithm was increased sensitivity to small fire, we used these relatively low thresholds in this test.

### 3.3. Random Forest

RF is widely used for classification and regression in various remote sensing applications [40–44]. RF is based on the CART (classification and regression tree) method [45], in which RF uses many independent decision trees for regression and classification using

(weighted) averaging and majority voting, respectively. RF uses two randomization strategies: random selection of training samples for each tree and random selection of input variables for each node of the tree [46–48]. By developing many independent trees from different sets of training samples and input variables, RF attempts to provide relatively unbiased results [49–51], preventing overfitting and sensitivity to the training data configuration.

The RF algorithm is fast, easy to parameterize, and robust [52,53]. In addition, it can quantify the importance of features, which makes it possible to use it for feature ranking or selection [54]. Many studies have shown that selecting a subset of features from an excessive number of feature variables is essential to prevent overfitting and reduce the computational complexity of the model [55–58]. Many studies [17,19,25] have been conducted to show that RF models are effective in fire detection. Therefore, we chose the RF model for fire detection.

The construction process for the fire detection RF models is shown in Figure 2, which consists of five key processes: (1) cloud and water masks, (2) multisource data matching including spatial and temporal dimensions, (3) feature selection, (4) optimization of model parameters, and (5) construction and validation of the model on the validation dataset.

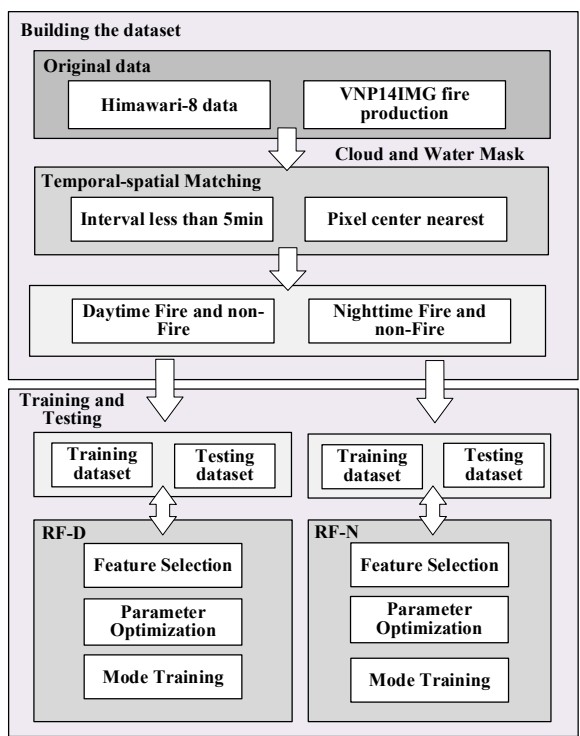

**Figure 2.** Flowchart for fire detection model construction.

### 3.3.1. Building the Dataset

In this study, we first constructed an a priori dataset to train and validate the fire detection model. Systematic bias between AHI and VIIRS had to be considered when constructing the training sample for the Himawari-8 fire detection model using the VIIRS fire product. In this study, we introduced the concept of fire pixel count: first, we regarded the Himawari-8 pixel as composed of $5 \times 5$ small grids of $0.004° \times 0.004°$; if the center of a VIIRS fire pixel fell into a small grid, we labeled the grid; and, finally, we took the number of labeled grids in the Himawari-8 pixel as the fire pixel count. We selected $0.004°$ as the small grid size because the Himawari-8 grid size was $0.02°$, and the resolution was about five times that of VIIRS. Then, we constructed the fire labels with different fire pixel count thresholds and trained the RF models separately. By comparing the performance of

different models on the validation dataset, we used the pixel with fire pixel counts greater than 7 (10) in Himawari-8 as the fire label dataset for the daytime (nighttime) model, and we selected the non-fire labels from the potential fire pixels according to 1:10. To reduce the effects of striping phenomenon and localization errors of the VIIRS fire product [59,60], we did not select fire labels adjacent to 8 pixels. In addition, because of the large spatial and temporal variability of fire, we retained only fire with a time difference of 5 min or less between VIIRS and AHI observations to construct the dataset.

We selected the November and December 2019 Himawari-8 data and VIIRS fire product to construct the dataset according to this method. To improve the generalization ability of the model, we used the stratified random sampling method to divide the data into the training dataset and validation dataset according to 7:3 for model construction. Stratified random sampling is a sampling method that divides the target object into homogeneous groups before using simple random sampling to select elements from each group to be a part of the sample group [61,62]. Considering the imbalance between the fire and non-fire labels, by dividing the dataset by the stratified random sampling method, we ensured the same ratio of fire and non-fire labels in the training and validation dataset, which effectively prevented the possibility of too few fire labels after dividing the unbalanced dataset. Finally, we collected the January 2020 dataset for testing.

### 3.3.2. Feature Selection

Feature selection is an important step in the parameter retrieval of ML algorithms. Feature selection can reduce the dimensionality of the training data, avoid overfitting, and improve the operation efficiency of the model. It also makes the model more explanatory [36]. First, based on the original data, we selected a total of 12 features, including thermal infrared brightness temperature (Tbb07–Tbb16), longitude, and latitude data. To achieve a consistent retrieval of fire pixel features, we excluded solar zenith angle data, which were strongly correlated with time. Referring to the existing studies [17,25] for the original bands, we calculated the radiance (Rad) and combined the radiance and brightness temperature bands separately to obtain 25 band calculated features. In addition, the spatial features of the pixels reflected the difference between the fire pixels and the background pixels. In this study, we extracted five spatial features with reference to the MODIS thermal anomaly algorithm [9]: the mean absolute deviation of the background pixel band ($\delta_{07}$, $\delta_{14}$, $\delta_{07-14}$), the difference between the pixels band, and the mean value of the background pixels band ($Tbb07 - \overline{T}_{07}$, $Dif_{07-14} - \overline{T}_{Dif_{07-14}}$). To analyze the relationship between different underlying surface types and the occurrence of fire, we also included land use data in the experiment. Table 2 summarizes the input variables used in the RF models.

**Table 2.** Features used to construct the RF model.

| | |
|---|---|
| Original band features | Tbb07, Tbb08, Tbb09, Tbb010, Tbb11, Tbb12, Tbb13, Tbb14, Tbb15, Tbb16, Lat, Lon |
| Band calculation features | Tbb07-Tbb11, Tbb07-Tbb12, Tbb07-Tbb13, Tbb07-Tbb14, Tbb07-Tbb15, Tbb12-Tbb16, Tbb13-Tbb14, Tbb13-Tbb15, Tbb07/Tbb09, Tbb07/Tbb10, Tbb07/Tbb11, Tbb07/Tbb12, Tbb07/Tbb13, Tbb07/Tbb14, Tbb07/Tbb15, Tbb07/Tbb16, Tbb09/Tbb16, Tbb13/Tbb15, Rad04-Rad07, Rad05-Rad07, Rad06-Rad07, Rad07-Rad12, Rad07-Rad15, Rad12-Rad15, Rad07 |
| Spatial features | $\delta_{07}$, $\delta_{14}$, $\delta_{07-14}$, $Tbb07 - \overline{T}_{07}$, $Dif_{07-14} - \overline{T}_{Dif_{07-14}}$ |
| Auxiliary data | MCD12Q1.006 |

We used recursive feature elimination [63] for feature selection to remove features with potential covariance, reduce the complexity of the model, and improve the generalization ability of the model. Using the algorithm, we first calculated the importance of features according to the mean accuracy descent method [53,64] and then evaluated the

importance of features by disrupting the ranking of features and calculating the change in the accuracy of the model on the validation dataset. Then, we ranked the importance of the obtained features and removed the features with the lowest importance. We calculated the accuracy factor of the model on the validation dataset and then repeated the process for the remaining features until the number of features reached the specified minimum number of features. Finally, we selected the combination of features with the highest accuracy factor for model construction. Considering the imbalance of the dataset, we selected the F1-score as the accuracy evaluation factor for feature selection [65].

### 3.3.3. Optimal Model Parameter Selection

Similar to feature selection, parameter optimization of ML is a key step to improve model accuracy. In our study, we examined the effects of three key parameters of the RF model, including the number of trees, the maximum tree depth, and the maximum feature selection, on fire detection accuracy. Each fire detection model was set using a combination of 100 hyperparameters, where the number of trees ranged from 80 to 100 at a step size of 5, the maximum depth ranged from 5 to 20 at a step size of 5, and the maximum feature selection ranged from 2 to 10 at a step size of 2. The F1-scores were each calculated using the validation dataset as evaluation factors for tuning and optimizing the RF model. The models with a high F1-score and simple tree structure will be selected. Finally, we set the number of trees in the daytime model (nighttime) to 95 (85), the maximum depth to 15 (10), and the maximum feature selection to 4 (8).

### 3.3.4. Accuracy Assessment

For the accuracy assessment of ML results, most studies construct error matrices to calculate some evaluation metrics, such as the overall accuracy, recall, precision, and Kappa coefficient [23,66–68]. In this study, we used the confusion matrix of classification for statistical accuracy assessment. Because of the possibility of an imbalance between fire and non-fire classes, accuracy can be misleading in an unbalanced dataset [69]. Therefore, we considered four accuracy criteria: precision ($P$), recall ($R$), F1-score ($F$), and overall accuracy ($OA$).

$$P = \frac{TP}{TP + FP}, \tag{6}$$

$$R = \frac{TP}{TP + FN}, \tag{7}$$

$$F = \frac{FP}{TP + FP}, \tag{8}$$

$$OA = \frac{TN + TP}{TP + FN + FP + TN}, \tag{9}$$

where true positive ($TP$) is the number of correct positive predictions; false positive ($FP$) is the number of incorrect positive predictions; false negative ($FN$) is the number of incorrect negative predictions; and true negative ($TN$) is the number of correct negative predictions. Precision is the proportion of correctly predicted fires to the total number of predicted fires, and recall is the proportion of correctly predicted fires to the total number of true fires. The F1-score is the harmonic mean of recall and precision, which is used to represent the model output. The value range is between 0 and 1, and the closer the value is to 1, the better the model output is. OA refers to the probability that the classifier predicts correctly on the validation dataset.

## 4. Results

### 4.1. Model Accuracy Assessment

We constructed the RF models based on the selected optimal features and parameters and trained them on the training dataset. The RF models had high accuracy on the validation dataset, with an F1-score of 94.35% and OA of 99.86% for the RF-D model and an F1-score of 90.37% and OA of 99.69% for the RF-N model. We further validated the accuracy of the constructed models with the validation dataset, as shown in Table 3. Both the RF-D and RF-N models had high recall, which indicated that both models could identify the vast majority of fires. The precision of the RF-D model was higher than that of the RF-N model. This may have been related to the fact that the nighttime fire features were not obvious, which caused the RF-N model to easily classify non-fire pixels as fire pixels. Overall, the F1-scores of both the RF-D and RF-N models were high, 90.04% and 83.16%, respectively, and in addition, the OA metrics were both close to 1. This result was mainly due to the large number of non-fire pixels in the validation dataset that were correctly identified.

**Table 3.** RF-D and RF-N models accuracy assessment results on the validation dataset.

| | **Error Matrix** | | **Recall/%** | **Precision/%** | **F1-Score/%** | **OA/%** |
|---|---|---|---|---|---|---|
| | Predicted non-fire | Predicted fire | | | | |
| RF-D | Reference non-fire　103,750 | 190 | 93.70 | 86.66 | 90.04 | 99.74 |
| | Reference fire　　　　83 | 1234 | | | | |
| | Predicted non-fire | Predicted fire | | | | |
| RF-N | Reference non-fire　93,799 | 364 | 88.70 | 78.27 | 83.16 | 99.44 |
| | Reference fire　　　167 | 1311 | | | | |

### 4.2. Wildfires Monitoring

To further analyze the generalization ability of the RF models, we selected the January 2020 wildfires in the study area for sample area testing, which included the following: We used the VNP14IMG fire product as the validation data and selected the Himawari-8 data closest to the transit moment of VIIRS to extract the fire location using the RF models constructed in this study. We selected the Himawari-8 L2WLF product for the same period as the comparison data, and finally, we compared the RF models' extraction results and the Himawari-8 L2WLF product with the VNP14IMG fire product. Figure 3 shows the fire pixel extraction results of the different models and products for four wildfires.

In the daytime wildfire observation Himawari-8 images shown in Figure 3(a-1,b-1), the greenish-black area is the forest cover area, which shows the obvious columnar smoke from forest burning. The distribution of fire pixels is relatively scattered and mostly appears at the edge of the smoke, as shown in the corresponding VNP14IMG image. The yellowish and yellow areas in the nighttime wildfire observation Himawari-8 image shown in Figure 3(c-1,d-1) are the pixels with higher bright temperatures in band 7 in the image. The fire detected by the three algorithms appeared basically around these pixels. The Himawari-8 L2WLF product and RF models detected the fire pixels in the same area as VNP14IMG did, but both algorithms appeared to judge the area around the VNP14IMG fire pixel as a fire pixel. Compared with the Himawari-8 L2WLF product, the RF models were able to detect more fires, such as those in the green circle areas shown in Figure 3(a-2,b-2,c-2). Referring to the Himawari-8 L2WLF product description document [70], Himawari-8 L2WLF pixels are detected based on the normalized deviation of the 3.9 μm brightness temperature from the background temperature determined from the 10.8 μm brightness temperatures at surrounding 11 × 11 grids. We counted the Tbb07 and Tbb07-Tbb14 feature values of these fire pixels that the Himawari-8 L2WLF product failed to identify, and the results were all lower than the mean of the corresponding values of the reference fire pixels in each sample area. These pixels were missed by the Himawari-8

L2WLF product likely because the Himawari-8 L2WLF product is a threshold algorithm based on the Tbb07 and Tbb14 bands, and a higher threshold setting would have caused the algorithm to miss the detection. Sample area 4 shows that the Himawari-8 L2WLF product detected areas with higher bright temperature values, whereas RF-N detected areas with lower bright temperature values.

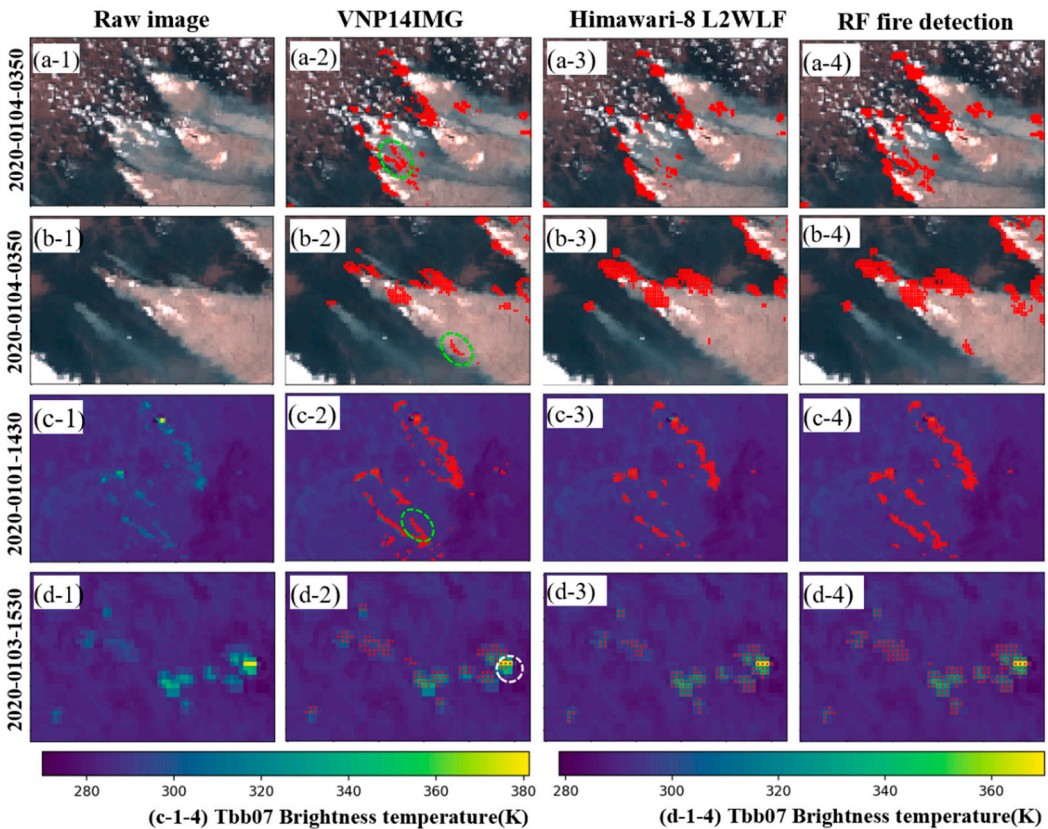

**Figure 3.** Fire detection results (column 1 is the original Himawari-8 image, where (**a-1,b-1**) are true color-composite images of bands 1, 2, and 3; (**c-1,d-1**) are single-band images of band 7; and columns 2, 3, and 4 are the results of fire pixels obtained from the different products and models overlayed onto the original image, where (**a-2,b-2,c-2,d-2**) are the fire detection results of the VNP14IMG product in four sample areas respectively; (**a-3,b-3,c-3,d-3**) are the fire detection results of the Himawari-8 L2WLF product in four sample areas respectively; (**a-4,b-4,c-4,d-4**) are the fire detection results of the RF models in four sample areas respectively ).

To quantify the detection accuracy of each model and product, we calculated the accuracy metrics of the different models and products, as shown in Table 4. The results showed that the Himawari-8 L2WLF product and the RF models had high recall for fire pixels with higher fire pixel counts, 85.64% and 95.62%, respectively, but both had relatively low accuracy rates, 64.18% and 59%, respectively. Compared with the Himawari-8 L2WLF product, the RF models had a slight decrease in precision (5.18%), but a significant increase in recall. In particular, for fire pixels with low fire pixel counts, the RF models had a 19.44% increase in recall. Compared with the RF-N model, the RF-D model had a high recall for fire pixels. Zhang et al. [25] used a similar random forest algorithm to detect wildfires in southwestern China. Compared with the study of Zhang et al., the method used in this paper had higher recall, which was mainly due to the ability to detect small fires better. This was mainly because previous studies only used VNP14IMG fire pixels with high confidence to construct a label dataset, while this paper added VNP14IMG fire pixels with nominal confidence to it and considered the representativeness issue.

**Table 4.** RF models and Himawari-8 L2WLF product accuracy evaluation on the test sample area.

| Sample Area | Himawari-8 L2WLF Product | | | RF Models | | |
|---|---|---|---|---|---|---|
| | Recall/% | Recall*/% | Precision/% | Recall/% | Recall*/% | Precision/% |
| 1 | 80 | 45.26 | 66.29 | 98.57 | 82.46 | 54.43 |
| 2 | 88.96 | 75.12 | 56.45 | 99.39 | 90.91 | 53.58 |
| 3 | 87.23 | 41.44 | 70.09 | 93.62 | 56.65 | 68.68 |
| 4 | 86.36 | 60.64 | 63.87 | 90.91 | 70.21 | 59.31 |
| Average value | 85.64 | 55.62 | 64.18 | 95.62 | 75.06 | 59 |

Recall*: Number of fire pixels with model detection fire pixel count below the threshold/Total number of fire pixels with fire pixel count below the threshold.

Notably, the white circles shown in Figure 3(d-2) are pixels with high bright temperature values, which are not recognized as fire pixels by VNP14IMG. Both the Himawari-8 L2WFL product and the RF-N model, however, recognized this part of the area as fire pixels. These may have been the fire pixels missed by the VNP14IMG algorithm, which indicated that the actual accuracy of the Himawari-8 L2WFL product and RF models may have had higher actual accuracy than the calculated values.

*4.3. Variable Importance Assessment in RF Classification*

After the feature selection of the 43 features, we used 25 and 23 features to construct the RF-D and RF-N models, respectively. The importance of the features in the model was ranked, and Figure 4 shows the importance scores of each feature of the model. The figure shows that the most significant factor affecting daytime fire detection was $\mathrm{Dif}_{07-14} - \overline{\mathrm{T}}_{\mathrm{Dif}_{07-14}}$, which was due to the fact that the peak wavelength of radiation shifted to the short wave when biomass was burned. As a result, the Tbb07 and Tbb14 bright temperature difference of the fire pixels was significantly enhanced compared with the neighboring pixels. The most important feature for nighttime fire detection was Tbb07/Tbb12, and this phenomenon also could be explained by fire-induced radiation variation. This feature was more important than other ratio features, which may have been related to the intensity of biomass burning in the study area. The top five features of both RF models were related to Tbb07, which indicated that fire detection depended heavily on Tbb07. Compared with the RF-D model, most spatial features were eliminated in the RF-N model feature selection. The experiments showed that although spatial features increased the recall of the RF models, the inclusion of spatial features increased the commission errors of the RF models because of the heating of the neighboring pixels by the fire pixels. At nighttime, however, the lack of some auxiliary information provided by light led to the reduction of the F1-score of the RF-N model on the validation dataset, and the spatial features were eliminated in the nighttime fire detection. In addition, the importance of land use features in both models was less than 0.5%, which indicated that coarse resolution underlying surface features had little effect when detecting fire using Himawari-8 data at large spatial scales.

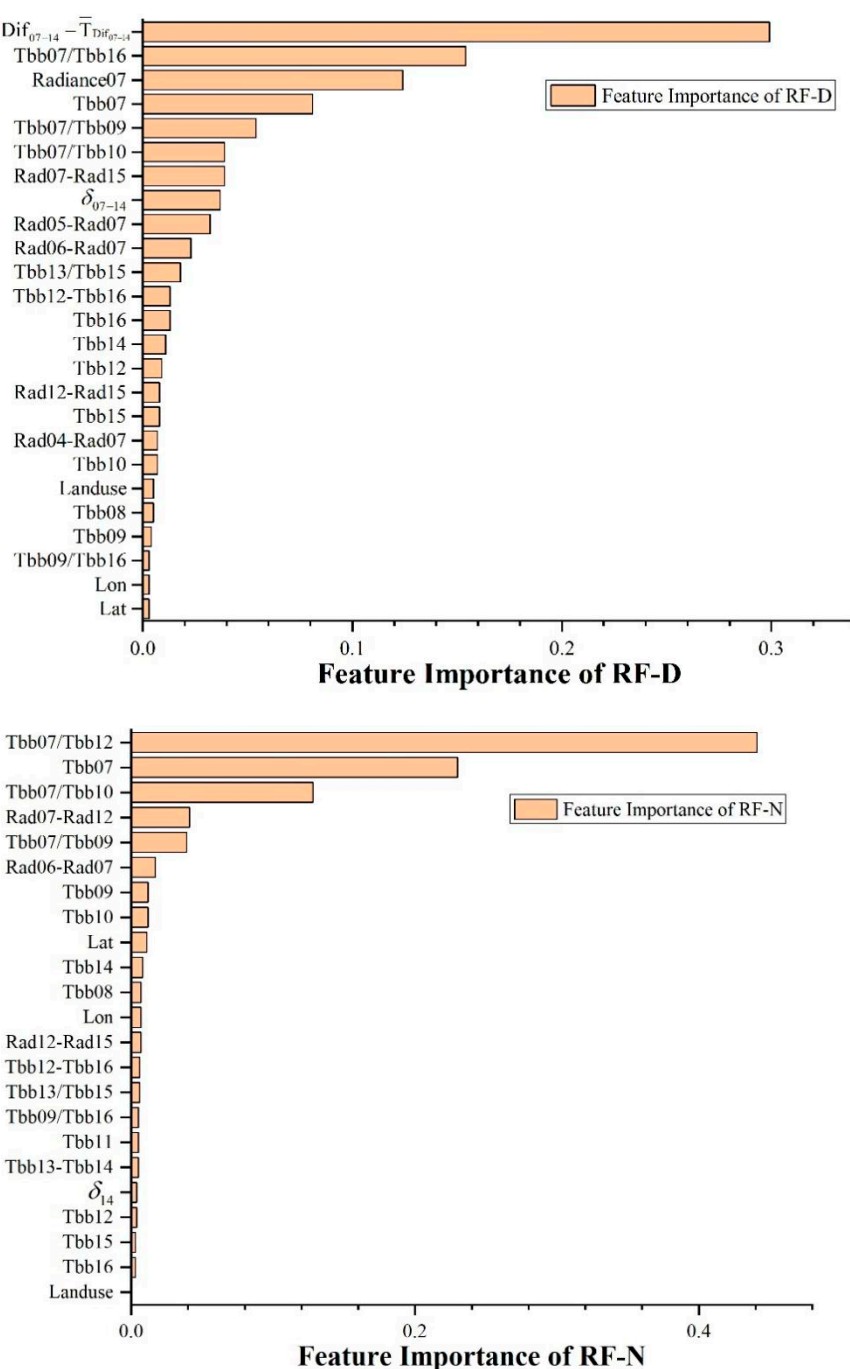

**Figure 4.** Ranking the features importance of RF-D and RF-N models.

## 5. Discussion

### 5.1. Pixel Representation

In this study, when constructing the RF model for fire detection, we evaluated the construction of a fire label dataset from the VIIRS fire product. Because of the uncertainty of the detection capability of Himawari-8 and VIIRS for the same fire, in this study, we first constructed fire labels with the threshold value of the number of VIIRS fire pixels within the Himawari-8 pixel, trained the RF models separately, and finally, selected a suitable fire label dataset by comparing the performances of different models on the validation dataset. Figure 5 shows the test results of the RF-D and RF-N models.

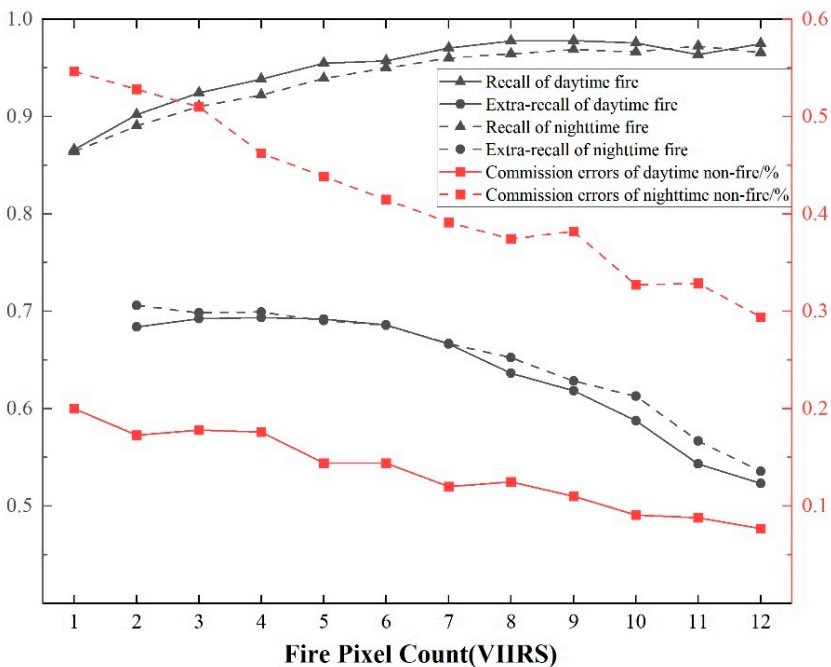

**Figure 5.** Accuracy of RF models on the validation dataset with fire label datasets constructed with different fire pixel counts as thresholds.

In Figure 5, the horizontal axis is the minimum threshold for the number of VIIRS fire pixels within the Himawari-8 fire label when training the RF models. The target recall is the recall of the trained model on the validation dataset for fire pixels above the threshold, the extra-recall is the recall of the model for fire pixels below the threshold, and the commission error probability is the probability of the model misclassifying non-fire pixels as fire pixels in the validation dataset. Overall, as the threshold increased, the target recall of the model increased, and the probability of commission error and extra-recall decreased. Although the model was better able to distinguish between large fires and non-fires, the detection ability for small fires decreased. In addition, the target recall and extra-recall of the RF-D model generally were consistent with the RF-N model, but the commission error of the RF-N model was two to four times higher than that of the RF-D model, which may have been related to the low threshold set by VIIRS for nighttime fire detection. These VIIRS-detected fire pixels may not have been represented as fire pixel information on the Himawari-8 images. Although using these pixels as labels eventually would enhance the ability of the model to detect small fires, it also would lead to a large commission error. In this study, we considered the 3 metrics and selected 7 and 10 as the thresholds for the RF-D and RF-N models, respectively.

### 5.2. Feature Selection

To analyze the effect of different input features on model accuracy, we trained the RF models with the original band information and all features. We obtained the accuracy of three different feature combination methods on the validation and test datasets, as shown in Table 5.

The overfitting phenomenon existed in different feature combinations, and the accuracy of the model on the test dataset was lower than that on the validation dataset. The overfitting of the RF models constructed using only the original band information was the most serious, and the F1-scores of the RF-D model and RF-N model decreased by 12.56% and 7.43%, respectively. This result indicated that adding band calculation features and the spatial features could reduce the overfitting of the model. Compared with the RF models constructed with all features, the RF models constructed with feature selection had a high F1-score, which was mainly reflected in the improvement of the precision; however,

the recall of the model decreased after feature selection. Compared with the RF-D model, the RF-N model had a decrease in accuracy, mainly in terms of precision.

**Table 5.** Accuracy of RF models constructed with different feature combinations on the validation and test datasets.

| | Feature Combination | Validation Set Accuracy | | | Test Set Accuracy | | |
|---|---|---|---|---|---|---|---|
| | | Recall | Precision | F1-Score | Recall | Precision | F1-Score |
| | Original feature | 94.95 | 93.97 | 94.46 | 91.57 | 74.16 | 81.95 |
| RF-D | All features | 96.84 | 91.83 | 94.27 | 94.76 | 81.88 | 87.85 |
| | After feature select | 96.43 | 92.35 | 94.35 | 93.70 | 86.66 | 90.04 |
| | Original feature | 96.39 | 81.82 | 88.51 | 93.23 | 71.73 | 81.08 |
| RF-N | All features | 96.51 | 82.32 | 88.85 | 92.49 | 75.23 | 82.97 |
| | After feature select | 95.57 | 85.69 | 90.36 | 88.70 | 78.26 | 83.15 |

### 5.3. Omission and Commission Error Analysis

Figure 6 shows the omission error of the RF models on the test dataset. As the number count of VIIRS fire pixels within the tested Himawari-8 fire pixels increased, the probability of omission of the Himawari-8 fire pixels all followed a decreasing trend. The RF-N model had a higher omission error than the RF-D model. For the Himawari-8 fire pixels with only one VIIRS fire pixel count, the probabilities of omission of the RF-D model and RF-N model were 59.81% and 75.9%, respectively. For the Himawari-8 fire pixels with 12 VIIRS fire pixels counts, all 92 fire pixels were detected during the daytime, whereas 23 out of 181 fire pixels were not detected during the nighttime.

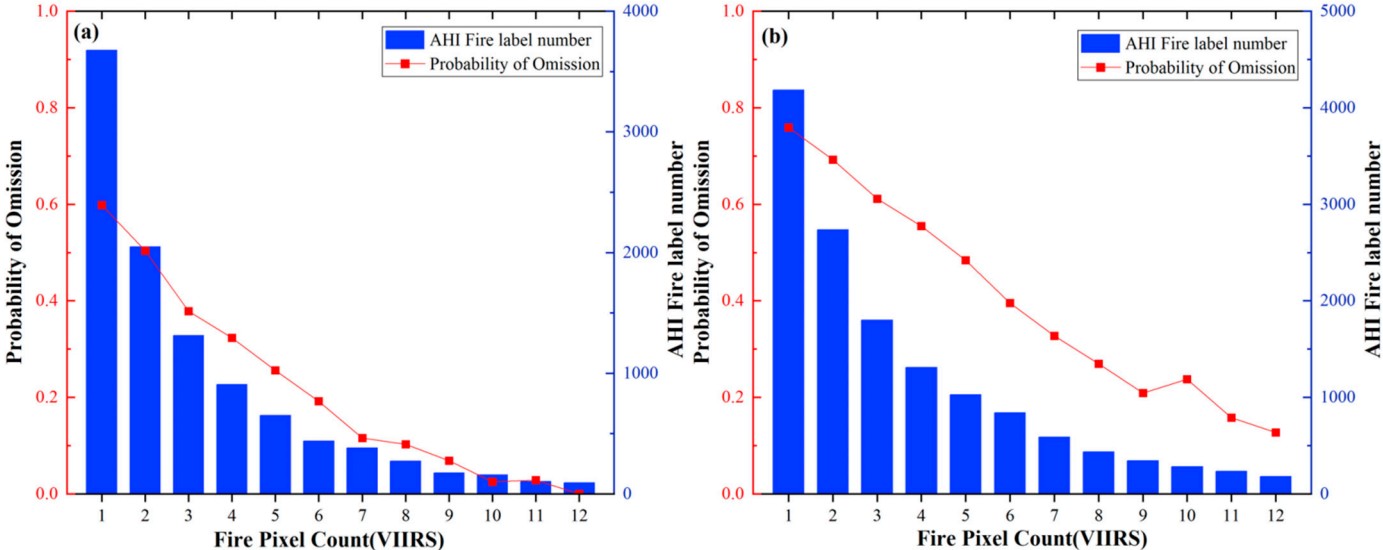

**Figure 6.** (**a**,**b**) show the probability of omission detections for the RF-D model and RF-N model, respectively, for Himawari-8 fire pixels containing different VIIRS fire pixel counts, and the histogram shows the total number of Himawari-8 fire pixels with different VIIRS fire pixel counts in the test dataset.

Figure 7 shows the probability of the distribution of VIIRS fire pixels in Himawari-8 pixels with different fire pixel counts. Figure 7a shows the case of only one VIIRS fire pixel, which appeared at the edges of the Himawari-8 pixel, most notably, at the four corners. The probability of appearing at the center of the pixel was smaller, and the probability of omission of these pixels was particularly high in Figure 6. As the VIIRS fire pixel count within the Himawari-8 pixel increased, the VIIRS fire pixel position gradually shifted

toward the center of the pixel, as shown in Figure 7c,d, and the probability of omission of the fire pixel gradually decreased. With a further increase of the fire pixel count, the VIIRS fire pixels were gradually and uniformly distributed within the Himawari-8 pixel. The actual spatial distribution of these fire pixels is depicted in Figure 8. Among them, the blue points were the Himawari-8 fire pixels with fire pixel counts 1–3, which were mostly at the edge of the fire clusters, or were independent fire pixels, and these pixels had a high probability of omission. Cyan points were Himawari-8 fire pixels with fire pixel counts 4–9, which were closer to the center of the fire clusters than the blue points and had a lower probability of omission. The red points were those with fire pixel counts greater than 10, which tended to be the center of the fire clusters, and they had a daytime and nighttime probability of omission of 0.014 and 0.113, respectively.

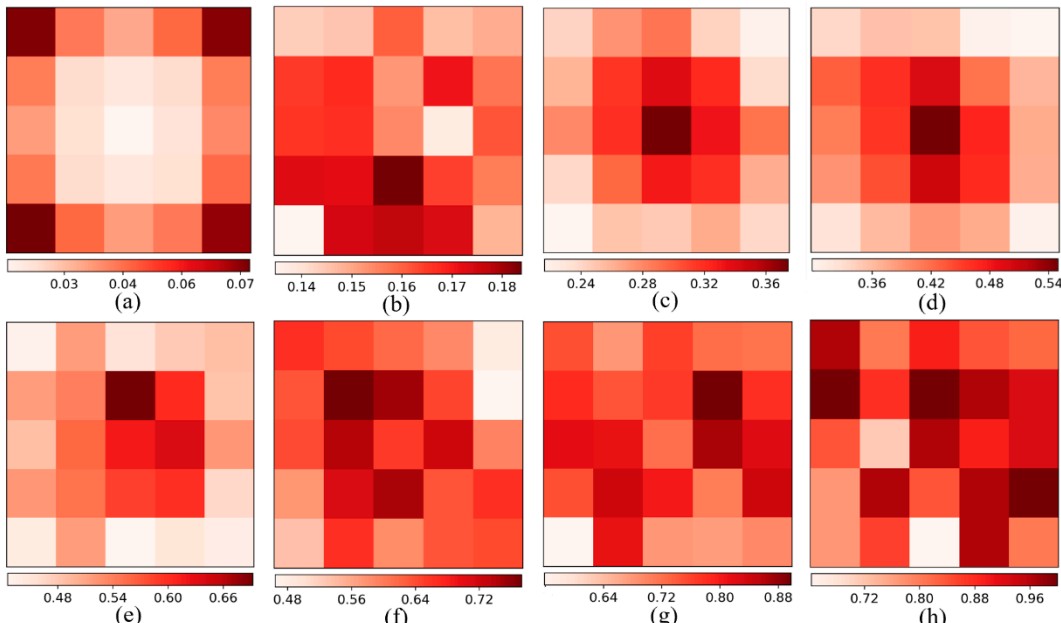

**Figure 7.** (**a**–**h**) show the probability of VIIRS fire pixel distribution among the Himawari-8 fire pixels for fire pixel counts 1, 4, 7, 10, 13, 16, 19, and 22, respectively. The probability of fire pixel distribution in the figure is obtained by adding the 5 × 5 0–1 matrix of each fire pixel count and dividing it by the total number of corresponding Himawari-8 fire pixels.

To further analyze the reasons for the probability of omission of these pixels, we counted the Tbb07 feature and the features of the most important of the detected and omission fire pixels under different fire pixel counting conditions, as shown in Figure 9. The small width of the box of each feature of the omission fire pixels and the fact that the upper quartile of the different features of the omission fire pixels was lower than the lower quartile of the detected fire pixels features indicated that the volatility of the eigenvalues of these omission fire pixels was small, although these features largely affected the detection of the fire. Except for the Tbb07/Tbb12 feature in Figure 9c, a large overlap existed between the detected and omitted fire pixels, which indicated that not one of the conditions was satisfied to be judged as a fire pixel. The enhancement of the eigenvalues of nighttime fires was not significant with the increase in fire pixel counts, which may be an important reason for the higher omission error of the RF-N model than the RF-D model.

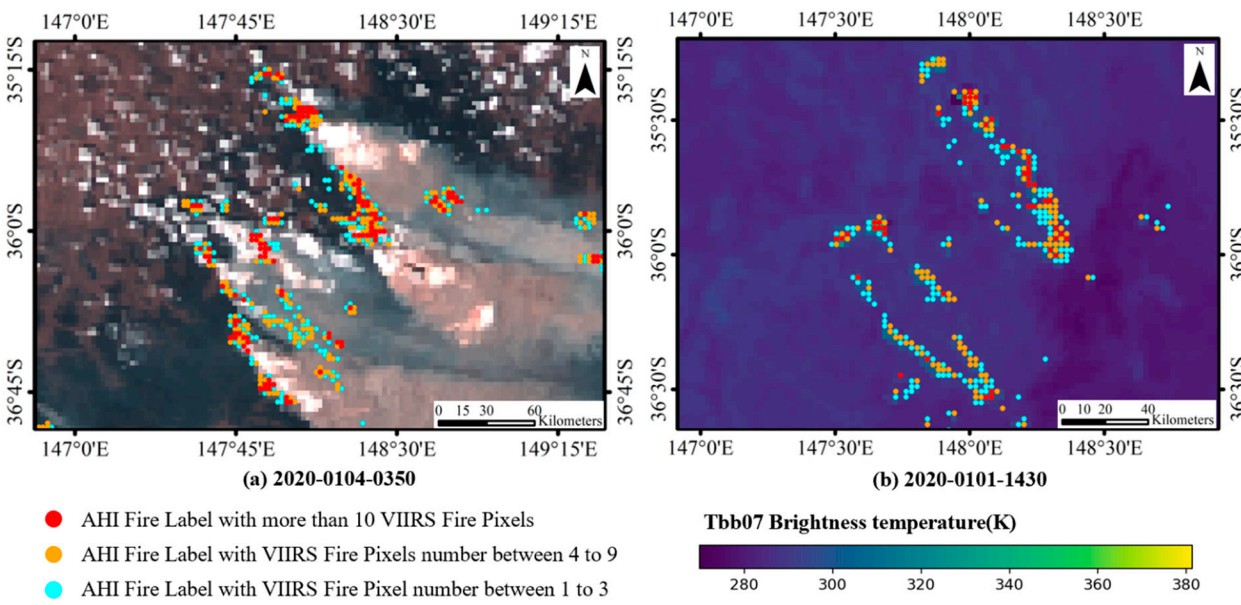

**Figure 8.** (**a**,**b**) show the spatial locations of Himawari-8 fire pixels with different fire pixel counts in sample area 1 and sample area 3, respectively.

**Figure 9.** Distribution of detection fire pixels and omission fire pixels for the RF-D and RF-N models for the first ranked feature and the Tbb07 feature for different fire pixel counts. (**a**,**b**) show the distributions of the $\mathrm{Dif}_{07-14} - \overline{\mathrm{T}}_{\mathrm{Dif}_{07-14}}$ feature and Tbb07 feature for daytime fire pixels, respectively;

(**c**,**d**) show the distributions of the Tbb07/Tbb12 feature and the Tbb07 feature for nighttime fire pixels, respectively. Note that there are no omission fire pixels at the daytime fire pixel count of 12.

For the commission error, Figure 10 depicts the distribution of the commission fire pixels in the sample areas. The commission pixels were distributed mainly near the fire clusters, which was similar to the results of Li et al. [71]. The commission of these non-fire pixels as fire pixels most likely was due to the occurrence of wildfires within the pixel, which could heat the ground in the adjacent area or produce large amounts of hot gas, which caused changes in the characteristics of the adjacent pixel. Figure 11 shows the distribution of the features of the commission fire pixel and detected fire pixels in the test dataset. In general, the boxes of the commission fire pixel eigenvalues were all located within the 1.5-times anomaly interval of the box of the detected fire pixel eigenvalues. This result indicated that these features likely caused the commission error of the fire pixels. Compared with the RF-D model, the median Tbb07 feature of the RF-N model commission fire pixels was closer to the detected fire pixels, which may explain the smaller precision of the RF-N model than that of the RF-D model.

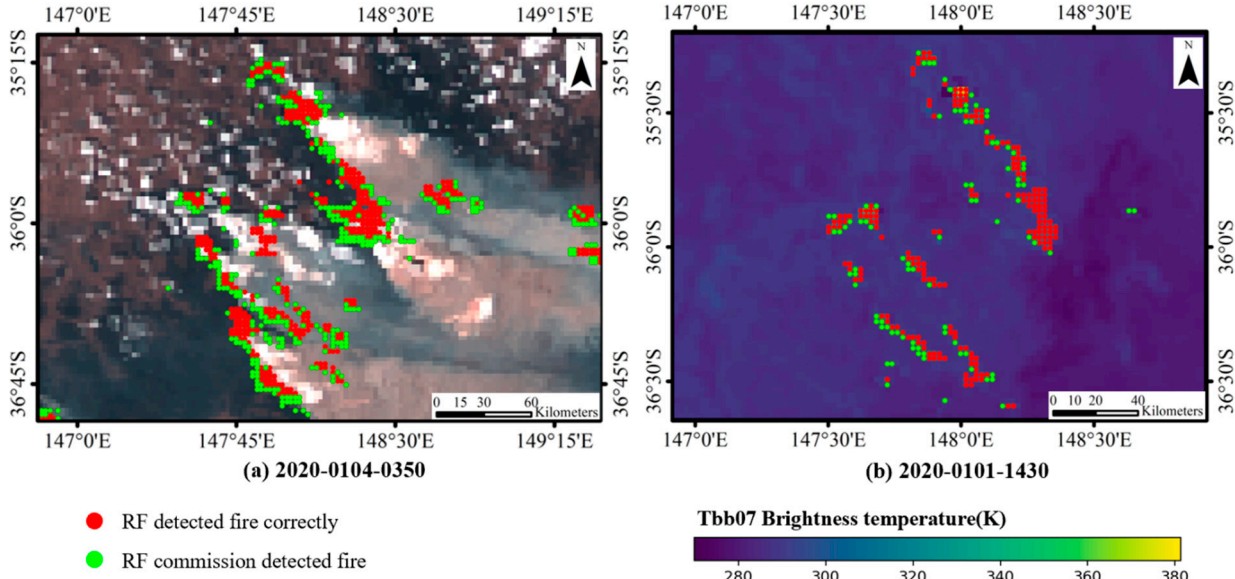

**Figure 10.** (**a**,**b**) show the spatial locations of the commission fire pixels in sample area 1 and sample area 3, respectively.

As can be seen in Figures 9 and 11, the omission and commission fire pixels produced by the RF fire detection model correspond to pixels with obscure and close features, respectively, indicating that these features were the main factors that caused the model to determine whether a pixel was a fire pixel or not. These omission and commission fire pixels were mainly distributed around the fire clusters. This might be due to the fact that the pixels located at the edge of the fire cluster have a small overfire area, resulting in less distinctive features of the pixels. Moreover, the burning fire pixels heat the ground in the adjacent area or generate a large amount of hot gas, resulting in enhanced features of the adjacent pixels.

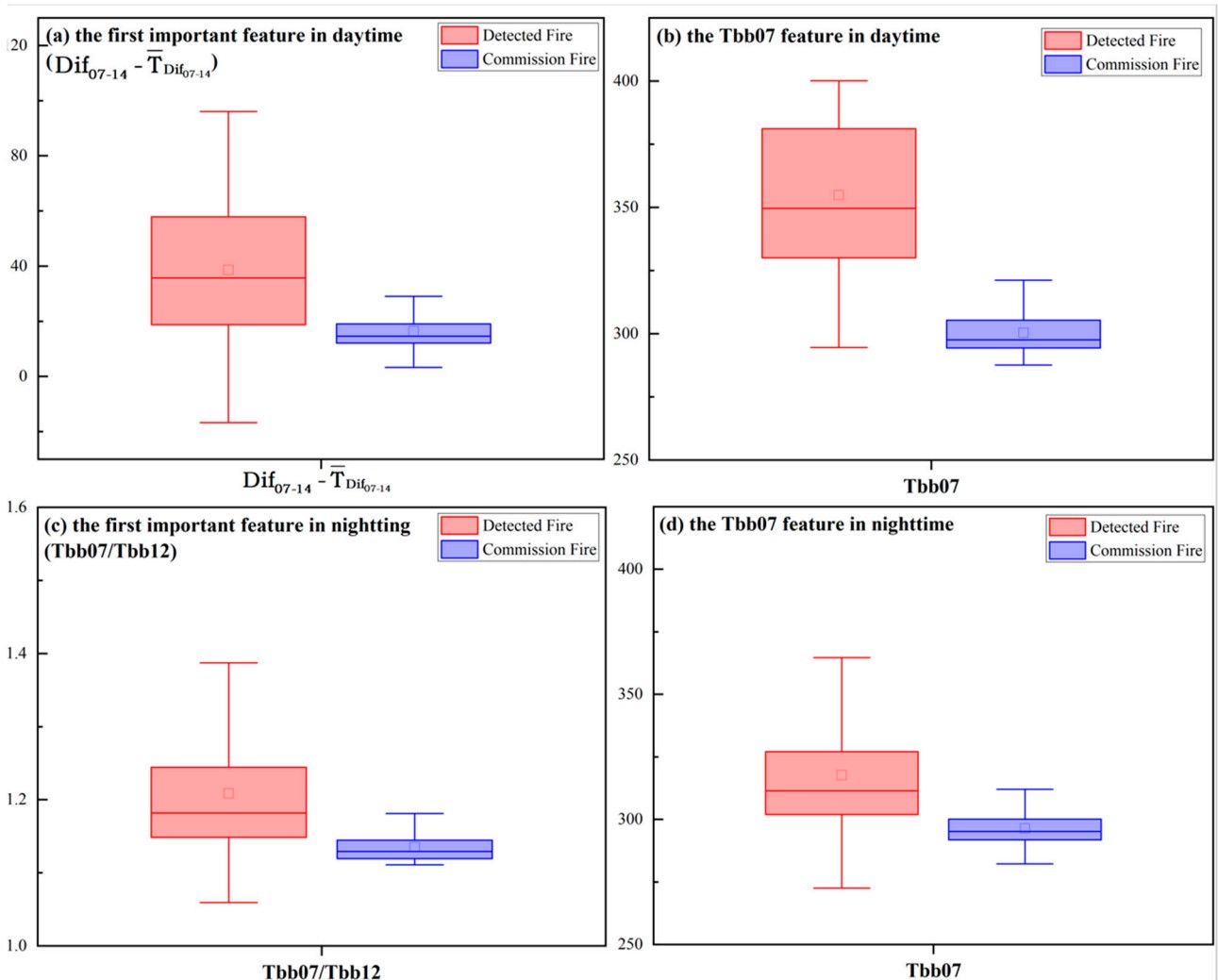

**Figure 11.** Distribution of detection and commission pixels for the first-ranked feature and Tbb07 feature in the RF-D and RF-N models. (**a**,**b**) show the distributions of the $\mathrm{Dif}_{07-14} - \overline{\mathrm{T}}_{\mathrm{Dif}_{07-14}}$ feature and the Tbb07 feature at daytime fire pixels, respectively; (**c**,**d**) show the distributions of the Tbb07/Tbb12 feature and Tbb07 feature at nighttime fire pixels, respectively.

## 6. Conclusions

In this study, we explored a combination of the VIIRS fire product and Himawari-8 data to achieve real-time monitoring of fire using RF models and feature selection methods. The fire detection model can monitor the fire location in real time and has excellent detection capability for small fires, making it highly significant for fire detection.

Compared with the Himawari-8 L2WLF product, the RF models used in this study had a high recall (95.62%). In particular, for fire pixels with low fire pixel counts, the model had a 19.44% improvement in recall.

Adding band calculation features, spatial features, and auxiliary data can improve the abundance of input information of RF models and then improve the accuracy of fire detection. The feature selection method effectively eliminated the redundant information caused by a large number of similar or correlated input features, reduced the possibility of overfitting, and thus, improved the accuracy of the RF models. The most important factors affecting daytime and nighttime fire detection are the $\mathrm{Dif}_{07-14} - \overline{\mathrm{T}}_{\mathrm{Dif}_{07-14}}$ feature and Tbb07/Tbb12 feature, respectively.

The fire detection accuracy of the RF-D model was higher than that of the RF-N model. This was mainly caused by the differences between fire and non-fire features, which were more obvious during the daytime than in the nighttime case. The omission

errors of the RF models occurred mainly in the detection of small fires, which were mostly located at the edges of the Himawari-8 pixel. In terms of spatial distribution, the omission errors and commission errors were concentrated primarily in the adjacent pixels of the fire clusters.

However, there are some limitations in this study. First, although the RF models have a high recall (95.62%), the precision of the models is low (59%). AHI has a high temporal resolution, and many studies have been conducted to detect fires by comparing the observed values of the bright temperature of the pixels with the predicted values based on historical observations. Future studies could consider adding temporal information of the brightness temperature of pixels to improve the precision of the model. Second, a wider variety of fire products are available for building fire labels. Meanwhile, the models need to be trained and tested globally. Third, the use of other novel deep learning architectures or the development of a new ad hoc model may also improve the achieved performance. Fourth, the fixed threshold reduced the accuracy of the cloud masking algorithm at different times and geographic locations, and thus, reduced the accuracy of fire detection. Future research could consider more generalizable methods for cloud masking, such as machine learning methods.

**Author Contributions:** Conceptualization, C.H. and D.Z.; Methodology, C.H., J.H., Y.Z., J.G., and D.Z.; Software, D.Z.; Formal analysis, C.H. and D.Z.; Investigation, C.H. and D.Z.; Writing—Original Draft Preparation, D.Z.; Writing—Review and Editing, C.H., W.H., P.D., Y.F., J.G., and D.Z. All authors have read and agreed to the published version of the manuscript.

**Funding:** This research was funded by the National Natural Science Foundation of China (Project No. 42130113), LZJTU EP 201806, and the Gansu Science and Technology Program (Project No. 22JR5RA090).

**Acknowledgments:** We are grateful for the use of the VNP14IMG fire product, Himawari-8 L1 grid data, Himawari-8 L2WLF product, and MCD12Q1.006 land use product in this study.

**Conflicts of Interest:** The authors declare no conflict of interest.

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
