# Peer review of "Real-Time Wildfire Detection Algorithm Based on VIIRS Fire Product and Himawari-8 Data"

_remotesensing, doi:10.3390/rs15061541_

Round 1
Reviewer 1 Report
This research article assessed the combined application of the high-spatial- 24 resolution product VNP14IMG and high-temporal-resolution Himawari-8 data in fire detection with a random forest model. To improve the quality of the paper, the authors need to consider the following comments for the revision.
Comments:
The author addressed real-time wildfire detection using random forest method and a combination of two-type data sets for better resolution and fast detection. So, the objective needs to be clearer.
The topic is not original in this filed. The author needs to mention the specific target based on the previous study.
Better resolution data and fast detection would be significance in this study, but author need to clarify.
The methodology used in this study seems not original, it’s from previous study (line-237-244). If it is original, more detailed explanation is necessary.
 What kind of data was used to compare the accuracy?
 The application of this study in other field can be checked.
 The formatting, units, figures can be revised and modification are needed.
It is suggested that the author revises English writing.
In abstract, it is advised to add some major findings from the review and some key drawbacks/scope remain for further study.
Introduction is well flown, need to clarify the objectives in a better way.
In materials and method, the description of Random forest model need to be a little detailed. Why author chose this method? How the model works with the current datasets. How the parameters were evaluated?
In results section, need to show some comparison with the previous study. The discussion with some reference study would make it clearer.
Conclusion is needed to be written with major finding of this article with drawbacks of the current method.
Reviewer 2 Report
In this manuscript, the purpose of this paper is to assess the combined application of the high-spatial- resolution product VNP14IMG and high-temporal-resolution Himawari-8 data in fire detection. This article summarizes several problems in the manuscript as follows.
1. The location of the study area in Figure 1 is not obvious enough. Please improve it.
2. Please add preprocessing of fire data (Himawari-8 data and VNP14IMG).
3. Please clarify the data fusion process of two different resolutions.
4. The discussion section lacks further discussion of the results obtained and the differences in accuracy with other data products or algorithms and the reasons.
Reviewer 3 Report
The manuscript titled "Real-Time Wildfire Detection Algorithm Based on VIIRS Fire Product and Himawari-8 Data" presents a sytudy for wildfire detection by using VIIRS product as well as using high temporal data from Himawari-8 satellite. The authors have developed an algorithm based on Random Forest classifier for mapping wildfire areas.
The introduction section needs to be strengthen more. Section 2, Section 3 and section 4 are well wrtitten. The authors should mention the limitation of their method clearly in discussion section.
Following are the detailed comments:-
1. There are already many papers has been published. what is new in this article ?
a. Zhang, Y., He, B., Kong, P., Xu, H., Zhang, Q., Quan, X., & Lai, G. (2021, July). Near Real-Time Wildfire Detection in Southwestern China Using Himawari-8 Data. In 2021 IEEE International Geoscience and Remote Sensing Symposium IGARSS (pp. 8416-8419). IEEE.
b. Xu, G., & Zhong, X. (2017). Real-time wildfire detection and tracking in Australia using geostationary satellite: Himawari-8. Remote Sensing Letters, 8(11), 1052-1061.
c. Jang, E., Kang, Y., Im, J., Lee, D. W., Yoon, J., & Kim, S. K. (2019). Detection and monitoring of forest fires using Himawari-8 geostationary satellite data in South Korea. Remote Sensing, 11(3), 271.
d. Kang, Y., Jang, E., Im, J., & Kwon, C. (2022). A deep learning model using geostationary satellite data for forest fire detection with reduced detection latency. GIScience & Remote Sensing, 59(1), 2019-2035.
e. Ding, C., Zhang, X., Chen, J., Ma, S., Lu, Y., & Han, W. (2022). Wildfire detection through deep learning based on Himawari-8 satellites platform. International Journal of Remote Sensing, 43(13), 5040-5058.
f. Wickramasinghe, C. H., Jones, S., Reinke, K., & Wallace, L. (2016). Development of a multi-spatial resolution approach to the surveillance of active fire lines using Himawari-8. Remote Sensing, 8(11), 932.
2. There is a major flaw in maps in the manuscript. The units of scale does not follow the SI conventions. Kilometres is always written as ‘km’ and not ‘KM’. Even while writing the full name the k will always be in smaller case.
3. L 193-201 please provide reference for the Satellite Product
4. What do you mean by thermal infrared brightness temperature (Tbb07–Tbb16). From where you got the values.
5. There is high resolution global LULC products like Copernicus Global Land Service: Land Cover 100m. why you are using MODIS course resolution LULC Data.
6. Has the methodology been used in other studies or analyses, or is it original? If it is original, further explanation is necessary.
7. I suggest finding a higher resolution land cover product for this analysis.
8. Which data has been used for accuracy assessment. I suggest to use burn area product to assess the accuracy.
9. It is suggested to check applicability of this techniques for cropland, shrubland, grassland and Forest fire.
10. Please add the band information about their spatial, spectral resolution in a table.
Round 2
Reviewer 2 Report
The author has made the modification carefully, and I suggest that it is acceptable.